# The Role of Alternative Lengthening of Telomeres Mechanism in Cancer: Translational and Therapeutic Implications

**DOI:** 10.3390/cancers12040949

**Published:** 2020-04-11

**Authors:** Marta Recagni, Joanna Bidzinska, Nadia Zaffaroni, Marco Folini

**Affiliations:** 1Molecular Pharmacology Unit, Department of Applied Research and Technological Development, Fondazione IRCCS Istituto Nazionale dei Tumori di Milano, 20133 Milano, Italy; marta.recagni@istitutotumori.mi.it (M.R.); nadia.zaffaroni@istitutotumori.mi.it (N.Z.); 22nd Department of Radiology, Medical University of Gdańsk, 80-214 Gdańsk, Poland; jbidzinska@uck.gda.pl

**Keywords:** adaptive response, ALT, resistance mechanisms, telomerase, telomerase inhibitors, telomeres

## Abstract

Telomere maintenance mechanisms (i.e., telomerase activity (TA) and the alternative lengthening of telomere (ALT) mechanism) contribute to tumorigenesis by providing unlimited proliferative capacity to cancer cells. Although the role of either telomere maintenance mechanisms seems to be equivalent in providing a limitless proliferative ability to tumor cells, the contribution of TA and ALT to the clinical outcome of patients may differ prominently. In addition, several strategies have been developed to interfere with TA in cancer, including Imetelstat that has been the first telomerase inhibitor tested in clinical trials. Conversely, the limited information available on the molecular underpinnings of ALT has hindered thus far the development of genuine ALT-targeting agents. Moreover, whether anti-telomerase therapies may be hampered or not by possible adaptive responses is still debatable. Nonetheless, it is plausible hypothesizing that treatment with telomerase inhibitors may exert selective pressure for the emergence of cancer cells that become resistant to treatment by activating the ALT mechanism. This notion, together with the evidence that both telomere maintenance mechanisms may coexist within the same tumor and may distinctly impinge on patients’ outcomes, suggests that ALT may exert an unexpected role in tumor biology that still needs to be fully elucidated.

## 1. Introduction

Telomeres are heterochromatic nucleoprotein complexes located at the termini of linear eukaryotic chromosomes [1]. The double-stranded DNA component of human telomeres comprises long tracts of 5′-TTAGGG-3′ tandem repeats, with the G-rich strand extending beyond its complement to form a 3′-overhang [1]. This single strand sequence may fold back onto itself and, by invading the double-stranded region of telomeric DNA, give rise to a lasso-like structure, referred to as telomeric (t)-loop, which creates a protective cap at the chromosomes ends [1]. Telomeric DNA and the t-loop are further stabilized by the shelterin complex, which plays a pivotal role in telomere protection and replication, hence in the regulation of telomere homeostasis [1]. This telomere-associated protein complex is composed of six individual factors. The three main components are the telomeric repeat binding factor 1 (TRF1) and 2 (TRF2), that bind to double-stranded telomeric DNA, and the protection of telomeres 1 (POT1), which binds directly to single-stranded telomeric sequences [1,2,3]. These proteins interact with additional factors, such as the TRF1-interacting protein 2 (TIN2), the ACD shelterin complex subunit and telomerase recruitment factor (ACD; formerly known as POT1- and TIN2-organizing protein, TPP1), and the transcriptional repressor/activator protein 1 (TERF2IP, also known as Rap1) [1,2,3].

Telomeres are crucial (i) to guarantee the genome integrity as they protect the chromosome ends from being recognized as DNA double-strand breaks and from unwanted DNA repair activities (i.e., non-homologous end-joining and homologous recombination) and (ii) to regulate the lifespan of cells [1,2]. Indeed, owing to the incomplete synthesis of lagging strand DNA (i.e., the end-replication problem [2]), normal/premalignant dividing (i.e., mortal) cells progressively lose telomeric sequences. This telomere shortening eventually results in dysfunctional uncapped telomeres hence in the improper chromosome end protection, which ultimately leads to the activation of a DNA damage response (DDR) and the consequent induction of senescence and/or apoptosis [1,2]. In this instance, telomere attrition acts as a tumor suppressor mechanism that imposes a barrier to cell proliferation, thus limiting cancer cell outgrowth [3]. Nonetheless, in cells lacking cell cycle checkpoints, dysfunctional telomeres may represent a source of genomic instability hence contributing to tumorigenesis [3]. The maintenance of telomere length and integrity has been described as an essential feature by which cancer cells attain replicative immortality (i.e., limitless lifespan) and stabilize their rearranged genomes [3,4].

## 2. Telomere Maintenance Mechanisms in Human Cancers

Two telomere maintenance mechanisms (TMM) have yet been identified in human cancers [5]. The most common TMM relies on the activation of telomerase, a reverse transcriptase (TERT) that, along with several accessory proteins [2], adds telomeric DNA to chromosome ends by using an RNA subunit (hTR) as a template for the de novo synthesis of telomeric repeats [6]. The accessory proteins (e.g., dyskerin; NOP10, NHP2, GAR1 ribonucleoproteins; reptin/pontin ATPases; telomerase associated protein 1 and WD repeat containing antisense to TP53) regulate the biogenesis, subcellular localization, and function in vivo of the telomerase holoenzyme [1,2,3,4,5,6].

Telomerase activity is not expressed in normal somatic cells, with the exception of few cell compartments (e.g., embryonic and multipotent stem cells, lineage progenitor cells) where it may become transiently activated and subsequently silenced upon differentiation [7]. Contrarily, telomerase activity is readily detectable in about 85–90% of human tumors, especially those of epithelial origin [5]. Hence, the possibility to interfere with its expression and/or function for therapeutic purposes has been actively pursued [4,5,6,7].

To maintain telomere length and integrity, human cancers lacking detectable telomerase activity (approx. 10–15%) may rely on the alternative telomere lengthening (ALT) mechanism [8], a homologous recombination (HR)-based process, which involves copying of telomeric DNA templates [9]. ALT-mediated TMM is commonly detected in tumors of mesenchymal and neuroepithelial origin [9,10,11], whereas it seems to be relatively rare in tumors of epithelial origin, although a consistent fraction of breast, gastric, and ovarian cancers, as well as malignant melanomas and neuroendocrine pancreatic tumors, have been reported to show ALT-associated features [10,11]. Moreover, it has also been reported that ALT activity may be a normal component of telomere biology in mammalian somatic cells in vivo [12]. This evidence suggests that ALT-dependent telomere maintenance may be present in normal somatic tissues, though at levels that are not sufficient to constrain telomere erosion and, similarly to telomerase, may become up-regulated in cancer [11,12].

Cells that maintain telomeres through ALT usually lack telomerase activity and display a combination of two or more molecular features [8,9,10], including (1) telomeres with heterogeneous length, which range from very short to more than 50 Kilobases; (2) the presence of a typical subset of promyelocytic leukemia (PML) nuclear bodies, referred as to ALT-associated PML bodies (APB), which contain telomeric chromatin and telomere- and HR-associated proteins; (3) high rate of telomere sister chromatid exchanges (T-SCE); (4) the presence of extrachromosomal telomeric DNA, which can be distinguished in double-stranded telomeric circles (t-circle), partial single-stranded circular DNA (G- or C-circles), linear double-stranded DNA and high molecular weight “t-complex”, that are thought to be products of t-loop resolution by recombination enzymes. The precise function of such an extrachromosomal telomeric DNA in telomere maintenance has not been clearly elucidated yet. However, C-circles are currently emerging as quantifiable markers of the ALT status [13,14]. It has indeed been reported that, on average, ALT cancer cells show ~1000-fold more C-circles than tumor cells with telomerase activity or non-immortalized cell lines [8]. A temporal correlation between the onset of ALT and the appearance of C-circles has also been observed in cultured cells that became immortalized spontaneously [8].

## 3. The Role of Telomere Maintenance Mechanisms in Tumors of Mesenchymal Origin

### 3.1. The Impact of ALT vs. Telomerase on Clinical Outcome

Although the activation of either TMM seems to be equivalent in supporting the limitless proliferative potential of tumor cells, the contribution of telomerase activity and ALT to tumor progression, hence to patients’ outcome, has been reported to differ prominently [15].

Tumors of mesenchymal origin, where ALT may be found in a significant proportion of lesions, represent an intriguing example within this frame (Table 1). One of the very first observations of the impact of either TMM on prognosis came from osteosarcoma (OS) patients [16]. In particular, it has been observed that the absence of any detectable TMM was associated with better overall survival of OS patients as well as that ALT-positive cases were as clinically aggressive as the lesions defined positive for telomerase activity, in terms of stage and clinical outcome [16]. Notwithstanding, it was also reported that TERT mRNA expression and telomerase activity were associated with unfavorable outcome in osteosarcoma, although TERT was the stronger of the two prognostic markers [17]. The evidence that TERT has a superior prognostic value than telomerase activity would suggest that its “extra-telomeric (anti-apoptotic/pro-survival) functions, which are independent of the telomere lengthening activity of the enzyme [18], may contribute to tumor aggressiveness. Furthermore, telomerase activity was found to be prognostic for disease relapse and cancer-related death in diffuse malignant peritoneal mesothelioma [19] and for cancer-related death in malignant peripheral nerve sheath tumor [20], whereas ALT failed to significantly affect clinical outcome in both cancer types (Table 1).

On the other hand, ALT has been reported to correlate with aggressive histological features and worse prognosis in leiomyosarcomas [21], as well as with an unfavorable prognosis in liposarcoma patients, both in univariable and multivariable analyses [23,24]. Furthermore, it has been reported that the presence of APBs was a significant prognostic factor for poor disease-free and overall survival at univariable analysis in uterine sarcomas [22], as well as that the ALT phenotype, defined on the basis of telomere length pattern or presence of APB, was associated with poorer survival in malignant fibrous histiocytoma [25,26] (Table 1).

Altogether, these findings suggest that the clinical impact of TMM is highly dependent on tumor type, with a seeming aptitude of ALT to adversely impact the outcome of soft tissue sarcoma patients [27]. In this context, it is worth mentioning that the ALT mechanism in cancer has been associated with complex karyotypes and chromosome instability [11] and, more recently, it is assumed to be influenced by genetic changes that may sustain its activation [28]. Specifically, a significant correlation between ALT activity and mutations or abnormal expression of ATRX chromatin remodeler and/or its binding partner H3.3-specific histone chaperone DAXX gene has been observed [9]. In particular, the loss of ATRX has been found to be highly associated with ALT-positivity in different subtypes of sarcomas with complex karyotypes [29], including dedifferentiated liposarcoma [24], angiosarcoma [30], and uterine smooth muscle tumors [31]. Moreover, the loss of ATRX or DAXX genes has been found to be associated with poor prognosis (i.e., death or recurrence) in smooth muscle tumors of uncertain malignant potential and early-stage uterine leiomyosarcoma [31].

This tangle is further complicated by the evidence that while telomerase activity may be unequivocally defined by the detection of its catalytic activity, sometimes combined with the assessment of the expression levels of TERT and its splicing variants, the occurrence of ALT in tumors may not be doubtless defined. At present, it is still based on the detection of one (often APBs) or a combination of two (APB and TRF) or more (APB, TRF, T-SCE, or C-circles) markers [11,15,32]. In this context, it has been documented that a significant fraction of lesions within a given tumor maintain telomeres in the absence of telomerase activity or detectable APBs [33]. These lesions may represent ALT tumors, where recombination-like activity at telomeres may occur in the absence of overt APBs [33]. Moreover, it has been reported that cells maintaining telomere length in the absence of telomerase activity and lacking usual phenotypic characteristics of ALT cells (i.e., APB, heterogeneous telomere length) may contain abundant C-circles [8]. This frame suggests that the APB screening may not be sufficient to correctly estimate the incidence of ALT in clinical tumors as well as that more than one ALT mechanism likely exists [11], thus highlighting that a universal consensus on which could be a reliable marker or a combination of markers that better define the ALT status of a tumor is still urgently needed. Furthermore, both TMM may coexist within a single tumor, though it remains unclear whether they are active in a single cell or the bulk tumor contains cell populations that use single TMM [11,15]. Moreover, it has been observed that TMM may undergo a shift (ALT↔Telomerase) in a cell cycle-dependent manner [34] as well as during disease progression, in that, an opposite TMM may operate in primary with respect to matched metastatic lesions [35]. These observations imply that the temporal shift of one TMM towards the other may confer a tumor with a distinct, or even opposite, biological behavior [19].

### 3.2. Preclinical Evidence of ALT Activation as an Adaptive Response to Telomerase Inhibition

It has been reported that telomerase-positive tumors often exhibit shorter telomeres than normal surrounding tissue counterparts [5], as a consequence of the late reactivation of telomerase activity during tumorigenesis [11]. This observation, alongside the evidence that the enzyme is expressed in about 90% of human tumors [5,11], have provided compelling arguments to indicate that the enzyme is a suitable cancer-associated target [36]. Data from preclinical studies on the effects of telomerase inhibition have led GRN163L (a.k.a. Imetelstat), a palmitoyl lipid-based thiophosphoramidate oligonucleotide inhibiting telomerase activity through competitive inhibition [36], to enter clinical trials as a single agent or in combination with conventional anticancer drugs for different malignancies (https://clinicaltrials.gov). Available data have demonstrated that the drug has limited therapeutic efficacy, though it was generally well-tolerated and with common side effects, such as neutropenia, thrombocytopenia, anemia, fatigue, etc. [36]. However, in brain tumor patients, the drug was administered for an average of only 2 weeks before toxicity occurred [37], and treatment interruption (i.e., medication vacation) led to rapid telomere lengthening and rescue of tumor growth [38]. On the other hand, Imetelstat has been reported to cause rapid and durable hematologic and molecular responses in essential thrombocythemia patients [39], and to be active in myelofibrosis, where it likely suppresses the proliferation of malignant progenitor cell clones, though with the potential to cause clinically significant myelosuppression [40].

An important issue, though poorly investigated yet, is the possible development of resistance to telomerase inhibitors [28]. Uncertainty still surrounds on whether or not anti-telomerase therapies will be hampered by the lag time needed for telomeres to erode at a critical length sufficient to induce tumor cell killing, and/or whether treatment-induced telomere crisis will result in the emergence of possible adaptive responses [28]. In this context, it can be reasoned that tumors relying on telomerase activity could become resistant to prolonged inhibition of telomerase activity [28], in that they may undergo a selective pressure, which allows the clonal expansion of cell sub-populations that may activate ALT as a compensatory mechanism [28].

Apparently, the abundance of preclinical data concerning the biological outcomes of telomerase inhibition would suggest that the emergence of ALT as an adaptive response to telomerase inhibitors is a relatively uncommon event [28]. Nonetheless, recent evidence alongside a new interpretation of pioneering experimental observations indicates that such an event may be less unlikely than previously expected. In support of this notion, is the very first evidence reported by Gan et al. [41], who showed that telomerase-positive human ovarian cancer cells exposed to the reverse transcriptase inhibitor AZT (3’-azido-deoxythymidine) for up to 9 weeks or to an antisense oligomer against human telomerase RNA (hTR) for up to 15 weeks, did not undergo telomere shortening despite an almost complete abrogation of the telomerase activity [41]. These telomerase-inhibited cells showed a relatively homogeneous telomeric signal without overt features of ALT (i.e., APBs or heterogeneous telomere length), evidence that was interpreted at that time as a clue for the existence of a telomerase-independent mechanism for telomere maintenance [41].

A step forward was made in 2004 by Betcher and colleagues [42], who showed for the first time that telomerase inhibition by a dominant-negative hTERT triggered a telomerase-independent, ALT-like telomere elongation mechanism in mismatch repair-deficient human colon cancer cells [42]. This study has contributed chiefly to pave the way toward the notion that ALT may become engaged in cancer cells exposed to telomerase inhibitors and act as an unsuspected mechanism of resistance to anti-telomerase therapy [42].

This notion was reinforced years later by the observation that T-cell lymphomas developing with high penetrance in *ATM*^−/−^ transgenic mice re-emerge with acquired features of ALT following an initial growth inhibition due to genetic extinction of telomerase [43]. Similarly, it has been reported that ALT mechanisms can be induced, both in terms of heterogeneous telomere length and high frequency of APBs, in hTERT-immortalized primary esophageal epithelial cells and genetically defined fully malignant transformed human keratinocytes upon inhibition of telomerase either following the infection with a mutant hTR-expressing lentivirus or the RNAi-mediated depletion of hTR as well as the combination of both approaches [44].

Moreover, it has been recently reported that ATRX and DAXX deletion results in an increased number of APBs and C-circle DNA production in HTC75 cancer cells characterized by partial suppression of telomerase activity and associated telomeric-specific DNA damage consequent to the overexpression of a deleted form of ACD shelterin complex subunit and telomerase recruitment factor, formerly known as TPP1 [45]. In addition, the subsequent CRISPR/Cas9-mediated hTERT knockout on this cell background allowed to select cell clones that, after 40 population doublings in culture, showed features of ALT. These included elongated and heterogeneous telomeres as those observed in the ALT-naïve U2-Os osteosarcoma cells [45]. Similarly, a significant increase in both APB number and T-SCE was observed in human laryngeal cancer cells upon RNAi-mediated depletion of TERT and consequent inhibition of telomerase activity [46].

Negligible information is thus far available on the possible role of miRNAs—small non-coding RNAs that negatively regulate gene expression [47]—in the regulation of ALT activity in cancer cells [15]. Nonetheless, it has been recently reported that the ectopic reconstitution of miR-380-5p expression levels in telomerase-positive malignant peritoneal mesothelioma cells induced a mild, though significant, reduction of cell growth over time. This event was associated with a remarkable inhibition of telomerase activity and a marked decrease in the expression levels of telomerase associated protein 1 (TEP1), a direct target of the miRNA responsible for the proper functioning of telomerase [48]. However, no evident telomere exhaustion was appreciable in these cells after 3 months of reiterated ectopic transfection of miR-380-5p [47,48]. Contrarily, a slight but significant increase in mean telomere length was observed in these long-term transfectants. These cells continued to grow, though at a significantly reduced rate compared to control cells, and were characterized by a significant reduction in the expression levels of ATRX and by increased levels of C-circle DNA [48].

A summary of the available findings on ALT activation as a possible adaptive response to telomerase inhibition is reported in Table 2.

Pieces of evidence have shown that the ALT mechanism is particularly responsive to telomeric epigenetic modifications [49]. Methionine restriction, which affects DNA methyltransferase (DNMT) activity, results in a rapid accumulation of APBs [50], whereas depletion of anti-silencing function 1a/1b histone chaperones, which coordinate the shuttling of histone proteins during DNA replication, or of histone deacetylase 5 (HDAC5), results in ALT induction (i.e., APB formation or telomere recombination) in telomerase-positive cancer cell lines [51,52]. Similarly, miRNAs are known to play an active role in epigenetics (e.g., miR-29 family; miR-101; miR-148a [47]) by finely tuning the expression of specific epigenetic-associated factors, such as DNMTs, HDAC, and Retinoblastoma-Like protein 2 (Rbl-2) [47], may be involved in the regulation of ALT, independently of telomerase activity. In this context, it has been reported that miRNAs belonging to the miR-290 family (i.e., miR-291-3p, miR-291-5p, miR-292-3p, miR-292-5p, miR-293, miR-294, and miR-295) are markedly down-regulated in Dicer-1-null mouse embryonic stem cells and their ectopic expression results in the repression of Rbl-2, a validated direct target of the miR-290 family [53]. Of note, Dicer-1-null cells showed reduced amounts of DNMT1, 3a and 3b, which are commonly repressed by Rbl-2, and are characterized by hypomethylation of genomic DNA, including the subtelomeric regions of chromosomes, and by an increase in telomere recombination events and aberrant telomere elongation, two typical ALT-associated features [53]. These findings support the hypothesis that defects in miRNA processing machinery, such as Dicer-1 deletion, affect the epigenetic status of subtelomeric regions, thus creating a permissive milieu that allows ALT-associated recombination activity to take place [53].

In this context, it has been reported that among epi-miRNAs (e.g., miRNAs that regulate epigenetic factors at post-transcriptional level), miR-148a-3p directly targets DNMT1 in cancer [47]. Our preliminary observations (unpublished data) indicate that telomerase-positive cancer cells have lower endogenous levels of the epi-miR-148a-3p [47] with respect to ALT-positive cancer cells (Figure 1a). This evidence would suggest that owing to its role in epigenetics, miR-148a-3p over-expression may contribute to create an “ALT permissive” milieu. Of note, the ectopic expression of miR-148-3p levels in the telomerase-positive A549 lung cancer cells by miRNA mimic transfection (Figure 1b) had no effect on A549 cell growth, though it significantly affected their migrating capabilities in vitro (Figure 1c). Such an effect was paralleled by a decrease in the amounts (Figure 1d) of the epigenetic factor DNMT1 and of TRF2 interacting protein (TERF2IP), which plays a role in protecting chromosome end from aberrant homologous recombination [54], as well as by a moderate increase in the production of C-circle DNA (Figure 1d), even though no inhibition of telomerase activity was observed.

## 4. Conclusions

The activation of a TMM, namely telomerase activity, and the ALT mechanism, is at least in part responsible for the limitless proliferative capacity of tumor cells [4,5]. Whereas telomerase may be seen as a readily controllable mechanism that has emerged late during the evolution of eukaryotic organisms, ALT may be perceived as a kind of an ancestral TMM, that may be partly lost as soon as telomerase appeared [9]. Nonetheless, the mechanism underlying the activation of one TMM over another still remains to be clearly elucidated, even if a genetic basis (e.g., TERT promoter mutations, ATRX mutations, telomeric variant repeats) for the activation of either mechanism in cancer is becoming recognized [55].

Since the evidence of its selective reactivation in most human tumors, telomerase has attracted much attention as a possible cancer-associated target, and several strategies aimed to interfere with its expression and functions therapeutically, have been widely documented [5,6]. By contrast, no genuine ALT targeting therapies have been developed thus far, due to the fragmentary information available on specific molecular factors involved in the engagement and maintenance of such a mechanism in human tumors.

Nonetheless, the classical view of TMMs as static properties of cancer, in that tumors adopting one TMM over the other during transformation maintain it indefinitely [56], has been challenged by the evidence that the two mechanisms coexist within a single tumor [57] and that they may undergo a shift during tumor progression (Figure 2) [23,35]. In this context, it has been suggested that tumor aggressiveness could be determined by the TMM dynamic shift rather than dictated by the specific TMM operating within the primary tumor [57]. This scenario resembles a biological condition known as “metastable” or EMT/MErT hybrid phenotype, a sort of dynamic status that contributes to the acquisition of distinct aggressive traits [58,59] and characterizes certain subtypes of mesenchymal tumors [58,59] where ALT is frequently detected [19,20,21,22,23]. This evidence underscores an unprecedented and thus far poorly characterized role for TMM dynamics in tumor biology.

In addition, the evidence that (i) ALT may operate as a backup mechanism when telomerase is inhibited (Table 2 and Figure 2), (ii) ALT-positive cells are characterized by higher levels of telomeric DNA damage than telomerase-positive cells [9], and (iii) different factors involved in DNA damage repair are required to promote ALT activity in immortalized cell lines [60], which suggests that ALT may dictate the sensitivity of tumor cells not only to telomerase inhibitors but also to radiation [14] and DNA damaging agents [61].

In conclusion, it is now emerging that tumors may not be simply considered as static entities with respect to the operating TMM. Instead, their biological significance still needs to be fully accounted for, as the TMM dynamic shift during the natural history of a tumor and the coexistence of both TMM within the same tumor cell population may have a profound, yet unforeseen, impact on patients’ outcome, both in terms of disease progression and response to treatments (Figure 2). Consequently, the development of robust assays for the correct assessment of ALT activity in human tumors and the acquisition of new knowledge on the molecular pathways responsible for TMM activation and for their reciprocal shift still represent two challenging issues to be addressed that are of paramount importance in the field of cancer telomere biology.

## Figures and Tables

**Figure 1 cancers-12-00949-f001:**
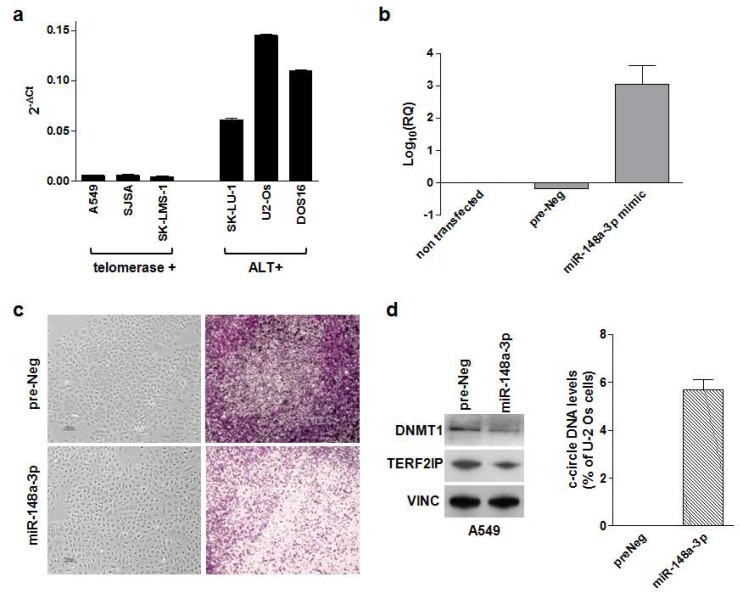
Effects of miR-148a-3p reconstituted expression levels in telomerase-positive A549 lung cancer cells: (**a**) Analysis of endogenous basal miR-148a-3p expression levels in a panel of telomerase-positive and ALT-positive lung cancer (A549, SK-LU-1); osteosarcoma (SJSA, U2-Os) and leiomyosarcoma (SK-LMS-1, DOS16) cell lines [10]. The endogenous levels of the miRNA were assessed by real-time RT-PCR. Data have been reported as 2^-ΔCt^ and represent mean values ± s.d.; (**b**) analysis of miR-148a-3p expression levels in A549 cells upon a 72 h transfection with the miRNA mimic or a control oligomer (pre-Neg). Data have been reported as Log_10_ (relative quantity, RQ) ± s.d.; (**c**) representative photomicrographs showing the growth (left) and migration (right) of A549 cells at 72 h after the transfection with the preNeg or the miRNA mimic oligomers. Original magnification: ×10; Scale bar: 100 μm. (**d**) representative Western immunoblotting (left) showing DNMT1 and TER2IP protein amounts in preNeg- and miR-148a-3p-transfected cells. Vinculin was used to ensure equal protein loading. Cropped images of selected proteins are shown. The graph (right) shows the quantification of C-circle DNA levels in A549 cells 72 h after the transfection with preNeg or miR-148a-3p mimic. Data have been reported as a percentage of c-circle DNA levels in each sample with respect to U-2 Os cells used as positive control and represent mean values ± s.d. Methodological details have been provided in Appendix A.

**Figure 2 cancers-12-00949-f002:**
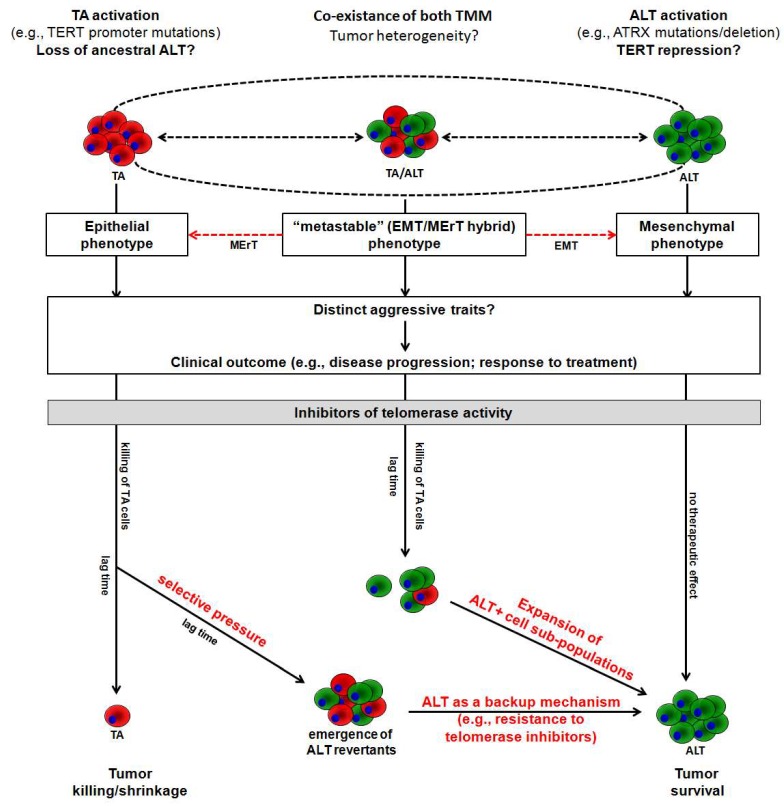
The dynamic shift (dashed black arrows) between telomere maintenance mechanisms during tumor progression resembles the “metastable” (EMT/MErT hybrid) phenotype described for a certain subtype of mesenchymal tumors. Such a dynamic condition may contribute to the acquisition of distinct aggressive traits and hence may differently affect the outcome of patients. In this frame, a plausible, though still hypothetical, scenario of the biological outcomes expected in tumors exposed to telomerase inhibitors has also been schematically reported as a function of the operating TMM. EMT, epithelial-to-mesenchymal transition; MErT, mesenchymal-to-epithelial reverse transition.

**Table 1 cancers-12-00949-t001:** Data reported by retrospective studies evaluating the impact of telomere maintenance mechanisms (TMM) on the clinical outcome of human cancer patients.

Tumor Type	Marker Analyzed	Clinical Endpoint	Outcome	Ref.
Osteosarcoma 71 lesions ALT frequency: 66%	TMM ^1^	OS	Patients with TMM-negative lesions showed a better outcome (*p* = 0.05), with 90% (95% CI, 71–100%) 5-year survival compared to 60% (95% CI, 45–76%) for patients whose tumors showed evidence of one or both TMM.	[16]
Osteosarcoma 44 lesions ALT frequency 79%	hTERT	DFS; OS	TERT-positive group showed a worse outcome compared to TERT-negative group. Three-year survival estimates were 21.4% ± 9.5% vs. 63.7% ± 11.1% (*p* = 0.014, DFS) and 42.9% ± 12.2% vs. 70.0 ± 9.9% (*p* = 0.031, OS), respectively.	[17]
	TERT/ALT ^2^	DFS	A significant difference (*p* = 0.012) in DFS was observed when the entire group of primary tumors was analyzed according to both TERT and ALT status. Three-year survival estimates were 50.0% ± 17.7% for TERT+/ALT+ patients (n = 6) and 62.3% ± 11.5% for TERT-/ ALT+ patients (n = 29). Patients with TERT+/ALT- lesions (n = 8) experienced disease relapse within 3 years. One patient with a TERT-/ALT- lesion treated with surgery only remained disease-free 13 years after diagnosis.	
	TA	DFS	Patients whose tumors had detectable TA (n = 5) experienced an unfavorable outcome compared with patients whose tumors lacked telomerase activity (n = 39). Three-year survival estimates were 20.0% ± 12.6% and 53.4% ± 9.7%, respectively (*p* = 0.073).	
Diffuse Malignant Peritoneal Mesothelioma 44 lesions ALT frequency 23%	TA/ALT ^3^	DFS; DRS	TA proved to be prognostic for the both endpoints (DFS: HR, 3.30; 95% CI, 1.23–8.86, *p* = 0.018; DRS: HR, 3.56; 95% CI, 1.03–12.5, *p* = 0.045), whereas ALT failed to significantly affect clinical outcome (DFS: HR, 0.40; 95% CI, 0.14-1.19, *p* = 0.10; DRS: HR, 0.45; 95% CI, 0.13–1.56; *p* = 0.21). In a subset of patients (n = 29) with resectable tumors who underwent cytoreductive surgery and hyperthermic intraperitoneal chemotherapy, TA proved to be prognostic for the DFS (HR, 3.32; 95% CI, 1.09–10.12; *p* = 0.03) and showed a trend towards an association with a poorer DRS (HR, 3.69; 95% CI, 0.79–17.13; *p* = 0.09). ALT did not affect clinical outcome.	[19]
Malignant Peripheral Nerve Sheath Tumors 57 lesions ALT frequency 37%	TA/ALT ^4^	DRS	TA proved to be prognostic for the endpoint (HR, 3.78; 95% CI, 0.79–17.13; *p* = 0.002), even when adjusted for the presence of NF1 syndrome and for margin status after surgical excision. Conversely, ALT status failed to affect clinical outcome, either using APB (HR, 1.25; 95% CI, 0.54–2.89; *p* = 0.61) or TRF analysis (HR, 0.57; 95% CI, 0.17–1.96; *p* = 0.38).	[20]
Uterine and Retroperitoneal/intra-abdomen Leiomyosarcoma 92 lesions ALT frequency 59%	ALT ^5^	OS	In univariate analysis, ALT phenotype was associated with a poor outcome (HR, 2.19; 95% CI, 1.10–4.34; *p* = 0.025) and showed to be an independent prognostic factor in multivariate analysis (HR, 2.67; 95% CI, 1.08–6.60; *p* = 0.034).	[21]
Uterine sarcoma 41 lesions ALT frequency 46%	ALT ^3^	DFS; OS	The presence of APB was a significant prognostic factor for poor DFS (*p* = 0.018) and OS (*p* = 0.021). The presence of APBs was not an independent prognostic factor in the multivariate analysis.	[22]
Liposarcoma 93 lesions ALT frequency 30%	TMM ^1^	DRS	At both univariable and multivariable analysis, TA alone did not prove to be associated with clinical outcome. ALT showed to be a prognostic indicator of unfavorable outcome both at univariable (HR, 2.70; 95% CI, 1.43–5.10; *p* = 0.0022) and multivariable (adjusted for tumor location, grade, and histology; HR, 3.58; 95% CI, 1.73–7.41; *p* = 0.0006) analyses. The presence of one or more TMM significantly (HR, 3.73; 95% CI, 1.76–7.88; *p* = 0.001) affected patient prognosis. Moreover, compared with TMM−cases, increased mortality was observed when TA and ALT phenotypes were considered separately with the TMM+ class, with adjusted hazard ratio estimates from the multivariable model of 2.58 (95% CI, 1.05–6.32; *p* = 0.0382) and 6.39 (95% CI, 2.64–15.49; *p* < 0.0001), respectively.	[23]
De-differentiated Liposarcoma 46 lesions ALT frequency 30%	ALT ^5^	DFS; OS	ALT phenotype was associated with adverse overall survival, albeit not statistically significant (HR, 1.954; *p* = 0.077). The marker was most significantly correlated with DFS (HR, 3.119; *p* = 0.003) compared with other clinic-pathological variable, such as mitotic count (HR, 2.689; *p* = 0.017), grade and stage (HR, 2.689; *p* = 0.017).	[24]
Malignant Fibrous Histiocytomas 43 lesions ALT frequency 33%	TA/ALT ^2^	OS	Univariate analysis revealed that ALT (HR, 0.367; 95% CI, 0.135–0.998; *p* = 0.0495) and TA (HR, 0.403; 95% CI, 0.147–1.107; *p* = 0.0779) were prognostic risk factors for death, though TA did not reach statistical significance. In the multivariate analysis ALT-positive status was the only independent prognostic factor for death (HR, 0.275; 95% CI, 0.104–0.724; *p* = 0.0089).	[25]
Bone Malignant Fibrous Histiocytomas 10 lesions ALT frequency 50%	TMM ^1^/TERT	OS	ALT-positive patients had a worse prognosis than other patients (survival rate, 20% vs. 80%, respectively, *p* = 0.0316) There was no significant correlation between the survival rate and the level of TA (*p* = 0.923) and of hTERT expression (*p* = 0.722).	[26]

^1^ TMM status defined on the presence of Telomerase activity, alternative lengthening of telomere (ALT) mechanism (defined by ALT-associated promyelocytic leukemia (PML) bodies (APB) detection and/or telomere restriction fragment (TRF) analysis) or both. TMM-negative specimens did not show any detectable TMM (nor TA or ALT); ^2^ ALT status was defined on the basis of TRF analysis; ^3^ ALT status defined on the basis of APB occurrence (tumor sections scored as APB positive if they contained APB in ≥0.5% of tumor cells); ^4^ ALT status defined on the basis of APB occurrence or telomere restriction fragment analysis; ^5^ ALT status defined on the basis of telomere fluorescent in situ hybridization analysis. Abbreviations: DFS, Disease-free Survival (time to relapse/recurrence or progression); DRS, Disease-related Survival; OS, Overall Survival; TA, Telomerase activity; TERT, Telomerase reverse transcriptase subunit; TMM, Telomere Maintenance Mechanism.

**Table 2 cancers-12-00949-t002:** Summary of the preclinical evidence showing the activation of the ALT mechanism as an adaptive response to telomerase inhibition.

Experimental Model	Telomerase Inhibitor	Outcome	Ref.
Human ovarian cancer cell line	AZT; antisense hTR	maintenance of homogeneous telomere length; no overt features of ALT	[41]
Mismatch repair-deficient human colon cancer cells	hTERT dominant negative	high molecular weight TRFs; T-SCE; low tumorigenic potential in nude mice	[42]
T-cell lymphomas developing in *ATM*^−/−^ transgenic mice	Genetic extinction of telomerase	increased heterogeneity in telomere length distribution; increased in APBs; occurrence of extrachromosomal telomere fragments	[43]
hTERT-immortalized primary esophageal epithelial cells; transformed human keratinocytes	mutant hTR-expressing lentiviruses; siRNA directed against hTR	heterogeneous telomere length; high frequency of APBs	[44]
Fibrosarcoma cell line expressing a deleted form of ACD and RNAi-mediated deletion of ATRX and DAXX	CRISPR/Cas9 knock-out of TERT	increased number of APBs; c-circle DNA production; elongated and heterogeneous telomeres	[45]
Human laryngeal cancer cell line	RNAi-mediated depletion of TERT	increased number of APBs; T-SCE	[46]
Human diffuse malignant peritoneal mesothelioma cells	miR-380-5p mimic transfection	slightly increased mean telomere length; reduced ATRX expression levels; occurrence of C-circles	[48]

Abbreviations: ACD, ACD shelterin complex subunit and telomerase recruitment factor (a.k.a. TIN2 interacting protein 1 or POT1 and TIN2 organizing protein); APB, ALT-associated promyelocytic leukemia body; ATM, ataxia telangiectasia mutated serine/threonine kinase; AZT, azidothymidine; ATRX, alpha-thalassemia/mental retardation syndrome X-linked chromatin remodeler; DAXX, death domain associated protein; hTERT, human telomerase reverse transcriptase; hTR, human telomerase RNA; RNAi, RNA interference; TRF, telomere restriction fragment; T-SCE, telomere-sister chromatid exchange.

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
