# Peer review of "The Role of Alternative Lengthening of Telomeres Mechanism in Cancer: Translational and Therapeutic Implications"

_cancers, 2020, doi:10.3390/cancers12040949_

Round 1

Reviewer 1 Report

This review summarizes the role of ALT in cancer and delineates the possible therapeutic implications.

The review is well written and includes the up to date knowledge in this field. I have only three minor points to pay attention to:

  1. I would detail the names of the shelterin components at the introduction, since the authors do relate to them later in the manuscript.
  2. Likewise, at line 59 I would spell out the rest of the telomerase accessories proteins in addition to dyskerin and WD repeats with antisense to TP53.
  3. At line 107- it is not clear how the authors came to the conclusion about telomerase extracurricular activities. Please explain.
  4. At the paragraph starts at line 177, it will be important to add that most tumors possess short telomeres than the normal surrounding tissue, due to the upregulation of telomerase as a late event. This adds to the prone and cones regarding telomerase inhibition as a therapeutic strategy.
  5. The paragraph regarding the authors preliminary results- should contain the rational behind focusing on epi miR 148a 3p.

Author Response

We are grateful to the Reviewer for the extremely positive evaluation of our manuscript. We welcome the useful tips she/he has provided to improve our manuscript.

Enclosed please find the point-by-point replies to the Reviewer’s comments:

This review summarizes the role of ALT in cancer and delineates the possible therapeutic implications.

The review is well written and includes the up to date knowledge in this field. I have only three minor points to pay attention to:

1) I would detail the names of the shelterin components at the introduction, since the authors do relate to them later in the manuscript.

The names of the six shelterin components have been detailed accordingly (lines 38-47).

2) Likewise, at line 59 I would spell out the rest of the telomerase accessories proteins in addition to dyskerin and WD repeats with antisense to TP53.

In addition to dyskerin and WD repeats with antisense to TP53, further telomerase accessory proteins have been quoted. As a consequence the original sentence has been changed accordingly (lines 65-70).

3) At line 107- it is not clear how the authors came to the conclusion about telomerase extracurricular activities. Please explain.

According to the reviewer’s suggestion, the text has been changed to make it more clear (lines 117-120). In particular, the observation that TERT expression levels was a stronger prognostic factor than telomerase activity has allowed hypothesizing that the “non canonical” anti-apoptotic/pro-survival functions of TERT, which are independent of the telomere lengthening activity of the enzyme, may contribute to tumor aggressiveness.

4) At the paragraph starts at line 177, it will be important to add that most tumors possess short telomeres than the normal surrounding tissue, due to the upregulation of telomerase as a late event. This adds to the prone and cones regarding telomerase inhibition as a therapeutic strategy.

The text has been modified accordingly (lines 179-184).

5) The paragraph regarding the authors preliminary results- should contain the rational behind focusing on epi miR 148a 3p.

The paragraph has been modified accordingly (line 273 and lines 287-299). Specifically, it has been reported that miRNAs may play an active role in epigenetics (e.g., miR-29 family; miR-101; miR-148a [see ref. 47]) by finely tuning the expression of specific epigenetic-associated factors, such as DNMTs, HDAC etc. In this context, the epi-miRNA miR-148a-3p has been documented to directly target DNMT1 in cancer [see ref 47]. Interestingly, we preliminary observed that telomerase-positive cancer cells have lower endogenous levels of the epi-miR-148a-3p with respect to ALT-positive cancer cells. This evidence allowed us hypothesizing that owing to its role in epigenetics miR-148a-3p over-expression may contribute to create an “ALT permissive” milieu. In particular, we found that the ectopic expression of miR-148-3p levels in the telomerase-positive A549 lung cancer cells significantly resulted in a decrease in the amounts of DNMT1 and of TRF2 interacting protein (TERF2IP), which plays a role in protecting chromosome end from aberrant homologous recombination. These events were paralleled by a moderate increase in the production of C-circle DNA, an ALT-associated marker.

Reviewer 2 Report

The authors comprehensively summarized current state of knowledge about the alternative lengthening of telomeres and described roles of alternative lengthening of telomeres in tumor biology. Additionally, the authors present their results showing effects of miR-148a-3p reconstituted expression levels in telomerase-positive A549 lung 275 cancer cells.

The manuscript is written very well, language is comprehensive, and text is structured logically.

I kindly suggest several points for the authors to improve the text and the presentation of results and mechanisms discussed.

1. Page 7, line 216: Instead “allowed select” should be “allowed to select” or “allowed selecting”.
2. Page 7, line 224-228: Rather long clauses should be divided into several shorter sentences.
3. Page 7, line 232-233: Rather long clauses should be divided into several shorter sentences.
4. Page 10, line 309-314, and line 316-320: The conclusions drawn in the two paragraphs would be more comprehensive and attractive for readers if authors provide them with a scheme summarizing the ideas in the conclusion part of the review e.g. mechanisms mentioned between lines 309-321.

Author Response

We are grateful to the Reviewer for the extremely positive evaluation of our manuscript. We welcome the useful tips she/he has provided to improve our manuscript.

Enclosed please find the point-by-point replies to the Reviewer’s comments:

The authors comprehensively summarized current state of knowledge about the alternative lengthening of telomeres and described roles of alternative lengthening of telomeres in tumor biology. Additionally, the authors present their results showing effects of miR-148a-3p reconstituted expression levels in telomerase-positive A549 lung 275 cancer cells.

The manuscript is written very well, language is comprehensive, and text is structured logically.

I kindly suggest several points for the authors to improve the text and the presentation of results and mechanisms discussed.

1) Page 7, line 216: Instead “allowed select” should be “allowed to select” or “allowed selecting”.

The error has been corrected (line 236).

2) Page 7, line 224-228: Rather long clauses should be divided into several shorter sentences.

The clauses have been modified accordingly (lines 237-242).

3) Page 7, line 232-233: Rather long clauses should be divided into several shorter sentences.

The clauses have been modified accordingly (lines 245-250).

4) Page 10, line 309-314, and line 316-320: The conclusions drawn in the two paragraphs would be more comprehensive and attractive for readers if authors provide them with a scheme summarizing the ideas in the conclusion part of the review e.g. mechanisms mentioned between lines 309-321.

According to the reviewer’s request, the ideas reported in the conclusion part of the manuscript have been schematically summarized in the new Figure 2.